# Effect of Catalyst Crystallinity on V-Based Selective Catalytic Reduction with Ammonia

**DOI:** 10.3390/nano11061452

**Published:** 2021-05-30

**Authors:** Min Seong Lee, Sun-I Kim, Myeung-jin Lee, Bora Ye, Taehyo Kim, Hong-Dae Kim, Jung Woo Lee, Duck Hyun Lee

**Affiliations:** 1Green Materials and Processes R&D Group, Korea Institute of Industrial Technology, Ulsan 44413, Korea; qneh5@kitech.re.kr (M.S.L.); sunikim@kitech.re.kr (S.-I.K.); leemj@kitech.re.kr (M.-j.L.); yebora@kitech.re.kr (B.Y.); thkim0215@kitech.re.kr (T.K.); hdkim@kitech.re.kr (H.-D.K.); 2Department of Materials Science & Engineering, Pusan National University, Busan 46241, Korea

**Keywords:** selective catalytic reduction, V-based catalyst, NO_X_ removal efficiency, crystallinity, isotropic heating

## Abstract

In this study, we synthesized V_2_O_5_-WO_3_/TiO_2_ catalysts with different crystallinities via one-sided and isotropic heating methods. We then investigated the effects of the catalysts’ crystallinity on their acidity, surface species, and catalytic performance through various analysis techniques and a fixed-bed reactor experiment. The isotropic heating method produced crystalline V_2_O_5_ and WO_3_, increasing the availability of both Brønsted and Lewis acid sites, while the one-sided method produced amorphous V_2_O_5_ and WO_3_. The crystalline structure of the two species significantly enhanced NO_2_ formation, causing more rapid selective catalytic reduction (SCR) reactions and greater catalyst reducibility for NO_X_ decomposition. This improved NO_X_ removal efficiency and N_2_ selectivity for a wider temperature range of 200 °C–450 °C. Additionally, the synthesized, crystalline catalysts exhibited good resistance to SO_2_, which is common in industrial flue gases. Through the results reported herein, this study may contribute to future studies on SCR catalysts and other catalyst systems.

## 1. Introduction

Air pollution has recently become a critical, global issue [1]. In response, environmental regulations have been tightened to reduce the emissions of chemical impurities (such as NO_X_, SO_x_, CO, volatile organic compounds (VOCs), and particulate matter (PM)) from power plants, boilers, and mobile sources [2,3]. Among the numerous air pollutants, nitrogen oxides (NO_X_: NO, NO_2_, and N_2_O) are extremely dangerous as they can easily disperse over long distances and form secondary PM_2.5_ by reacting with water vapor, which causes acid rain and smog, contributes to global warming [4,5], and can deeply penetrate human lungs, causing adverse health effects such as increased cardiovascular and respiratory morbidity [6].

Owing to the harmful effects of NOx, several technologies, such as selective catalytic reduction (SCR), selective noncatalytic reduction (SNCR), and non-selective catalytic reduction (NSCR), have been used to reduce NOx emissions. SCR with ammonia, which converts NO_X_ in fuel gas into N_2_ and H_2_O, is the most efficient NO_X_ removal technology, as the process emits no secondary pollutants and can reduce NOx emissions by 80–100% at a relatively low temperature (approximately 350 °C) [7,8].

Commercially, V_2_O_5_-WO_3_/TiO_2_ has been used as an SCR catalyst due to the strong catalytic activity of V_2_O_5_, and lower oxidation activity for the conversion of SO_2_ to SO_3_ in fuel gas. However, it has a narrow, high activation temperature range (300–400 °C), and its performance is reduced at low temperatures (below 300 °C), which induce the oxidation of SO_2_ to SO_3_ [9,10]. The flue gas temperature of industrial processes is typically as low as 300 °C, and the temperature of diesel engines has a wide range (100–400 °C) [5,11]. Therefore, the use of V_2_O_5_-WO_3_/TiO_2_ is restricted and requires adjustment, such as in the form of upstream installation and desulfurization.

Extensive studies have been conducted to develop new catalysts that can be effective under a low and wide temperature range of 200–450 °C. For example, Liu et al. designed a W-promoted MnO*_x_* catalyst (MnWO*_x_*) composed of a unique core–shell structure with Mn_3_O_4_ surrounded by Mn_5_O_8_, and achieved a high NOx reduction efficiency from 60 °C to 250 °C [12]. Huang et al. fabricated multi-walled carbon nanotube (CNT)-supported vanadium catalysts, in which vanadium particles were highly dispersed on the walls of the carbon nanotubes, which exhibited excellent activity in the SCR of NO at 100–250 °C [13]. However, the utilization of these catalysts in industrial fields is limited, as the catalysts are only activated at low temperatures, and they are deactivated when they come in to contact with the sulfur and water in exhaust gas at low temperatures below 300 °C [14,15].

The activity of catalytic materials is closely related to their crystalline structure [16,17]. Wang et al. reported that the formation of crystalline tungsten oxide on the surface of titania results in higher water resistance and NO_X_ removal efficiency at temperatures below 250 °C than those achieved by amorphous tungsten oxide [18]. Inomata et al. reported that V_2_O_5_ SCR catalysts with low crystallinity achieved better catalytic performance than that of V_2_O_5_ with high crystallinity [19]. They also reported that the crystalline V_2_O_5_ has a higher catalytic performance than amorphous V_2_O_5_ under the same sintering conditions [20]. Recently, many studies have been conducted on V-base SCR catalysts to enhance catalytic activity under low temperature [21,22]. However, the effect of catalyst crystallinity on the performance of V-based SCR catalysts remains unknown.

In this study, we explored the effect of crystallinity of V_2_O_5_-WO_3_/TiO_2_ on the NO_X_ removal efficiency and improved the catalytic performance under temperatures ranging from 200 °C to 450 °C by controlling the crystallinity, with excellent thermal stability. The crystallinity of the V_2_O_5_ and WO_3_ catalysts was adjusted by altering the heating methods, and this was evaluated via transmission electron microscopy (TEM), X-ray diffraction (XRD), Raman, and selected area electron diffraction (SAED) analyses.

## 2. Materials and Methods

### 2.1. Synthesis of V_2_O_5_-WO_3_/TiO_2_ Catalysts

Catalysts containing 2 wt.% and 10 wt.% of V_2_O_5_ and WO_3_/TiO_2_ were prepared following the impregnation method, respectively. NH_4_VO_3_ (0.256 g, 99.99%, Sigma-Aldrich Inc., St. Louis, MO, USA) and (NH_4_)_6_H_2_W_12_O_40_ xH_2_O (1.062 g, 99.99%, Sigma-Aldrich Inc., St. Louis, MO, USA) were dissolved in 100 mL of deionized water with oxalic acid (0.386 g, 99.999%, Sigma-Aldrich Inc., St. Louis, MO, USA), which acted as a solubility agent. TiO_2_ powder (8.800 g, NT-01, NANO Co., Ltd., Sang-ju, Republic of Korea) was mixed with the prepared solution, and the mixture was stirred for 2 h. The amorphous V_2_O_5_-WO_3_/TiO_2_ catalyst was prepared by drying the mixture on one side by heating the bottom of the beaker with a hot plate, while the crystalline V_2_O_5_-WO_3_/TiO_2_ catalyst was prepared following the isotropic heating method, in which the beaker was submerged in an oil bath. The prepared samples were dried at 110 °C for 12 h, and the obtained powders were then calcined at 500 °C in a furnace for 5 h under atmospheric pressure.

### 2.2. Catalyst Characterization

The surface morphology and elemental composition of the samples were investigated by field emission scanning electron microscopy (FE-SEM, model: SU8020/Hitachi, Tokyo, Japan), transmission electron microscope (TEM, model: JEM-2100F/JEOL Ltd., Tokyo, Japan), and electron energy loss spectroscopy (EELS) at an accelerating voltage of 10.0 kV. Additionally, we analyzed the extent of crystallinity using X-Ray Diffraction (XRD, model: Ultima IV/Rigaku, Tokyo, Japan) with Cu Kα (λ = 0.15406 nm) radiation in the 2θ range from 20° to 80° at a scan rate of 1°/min. The Raman spectra (Raman, model: alpha300s/WITec, Ulm, Germany) were measured using a 532 nm laser to generate an excited state to observe the structure of the catalysts. The textural properties were analyzed following the Brunauer–Emmett–Teller (BET, model: ASAP2020/Micromeritics Instrument Corp., Norcross, USA) method. X-ray photoelectron spectroscopy (XPS, model: K Alpha+/Thermo Scientific, Waltham, USA) was conducted with Al Kα radiation to confirm the oxidation states of the samples, and the binding energy of C1s was normalized as 284.8 eV. The reduction properties of the catalyst materials were measured by NH_3_-temperature-programmed desorption (NH_3_-TPD, model: AutoChem II 2920/Micromeritics Instrument Corp, Norcross, USA). The samples were pretreated at 150 °C in a current of N_2_ for 4 h to remove physiosorbed NH3 species and organic matters, and NH_3_ was then adsorbed with 10% NH_3_/He gas at 150 °C for 1 h. The TPD experiment was conducted under a temperature range of 100–900 °C. A H_2_-temperature-programmed reduction (H_2_-TPR, model: AutoChem II 2920/Micromeritics Instrument Corp, Norcross, USA) experiment was conducted, during which the samples were immersed in a current of 10% H_2_/Ar in the 150–900 °C temperature range.

### 2.3. Catalytic Measurement

The NH_3_-SCR activities were evaluated in a fixed-bed reactor under high atmospheric pressure. The operating temperature was varied from 200 °C to 500 °C, and the reactive gas was composed of 300 ppm each of NO, NH_3_ (NH_3_/NO_X_ = 1.0), SO_2_, and 5 vol.% of O_2_ with a balance of N_2_ at a total flow rate of 500 sccm. During evaluation, 0.35 mg of the powder catalyst (sieved to 40–60 mesh) was tested, which yielded a gas hourly space velocity (GHSV) of 60,000 h^−1^. The reactive gas concentration was continuously monitored via Fourier transform-infrared spectroscopy (model: CX-4000/Gasmet, Vantaa, Finland) and O_2_ analyzer (Oxitec 5000, Marienheide, Germany). The NO_X_ removal efficiency and N_2_ selectivity were calculated according to Equations (1) and (2), respectively.
(1)$${\mathrm{NO}}_{X}\;\mathrm{removal}\;\mathrm{efficiency}\;\left(\%\right)=\frac{{\mathrm{NO}}_{X\;\mathrm{inlet}}-{\mathrm{NO}}_{X\;\mathrm{outlet}}}{{\mathrm{NO}}_{X\;\mathrm{inlet}}}\times 100,$$
(2)$$N_{2}\;\mathrm{selectivity}\;\left(\%\right)=1-\frac{2N_{2}O_{\mathrm{outlet}}}{{\mathrm{NO}}_{X\;\mathrm{inlet}}+{\mathrm{NH}}_{3\;\mathrm{inlet}}-{\;\mathrm{NO}}_{X\;\mathrm{outlet}}-{\mathrm{NH}}_{3\;\mathrm{outlet}}}\times 100\;,$$

## 3. Results and Discussion

Figure 1 shows the SEM (a, b) and TEM (c, d) images of the V_2_O_5_-WO_3_/TiO_2_ catalysts prepared following the one-sided heating (a–c) and isotropic heating (b–d) methods. Both prepared V_2_O_5_-WO_3_/TiO_2_ catalysts exhibited similar particle sizes, shapes with diameters ranging from 15 nm to 50 nm, specific surface areas, pore volumes, and pore sizes (Table 1). The catalyst particles were composed of V_2_O_5_ and WO_3_ nanoparticles on TiO_2_ supports. It should be noted that the prepared nanoparticles were similar in size to the TiO_2_ powders [23]. The insets of Figure 1c,d show the EELS elemental mapping of the prepared catalysts, in which the red, blue, and green areas indicate V, W, and Ti, respectively. The V_2_O_5_ and WO_3_ were uniformly distributed on the TiO_2_ supports with no agglomeration, confirming that the drying process did not affect the morphology of the prepared catalysts. Table 2 shows the V_2_O_5_, WO_3_, and TiO_2_ weight fractions of the catalysts, respectively.

XRD measurements were taken to analyze the impact of the heating method on the crystalline structures of the prepared catalysts (Figure 2a). The prepared samples exhibit clear anatase TiO_2_ signals, V_2_O_5_ and WO_3_ signals were not observed because they are spread uniformly with low concentration [24]. Raman analysis was also conducted to determine how the heating conditions affected the structure of the V_2_O_5_-WO_3_/TiO_2_, as shown in Figure 2b,c. The spectra of the V_2_O_5_-WO_3_/TiO_2_ catalysts contained peaks at 144.7, 197.3, 401.5, 518.5, and 639.1 cm^−1^, in the spectra, which are typical of anatase TiO_2_ (Figure 2b) [25]. Figure 2c shows the structure of the vanadium and tungsten oxides in the 700–1100 cm^−1^ range. As active sites of V_2_O_5_-WO_3_/TiO_2_ catalysts in SCR reactions, the state of the vanadium oxide species on the surface of the V_2_O_5_-WO_3_/TiO_2_ plays a key role in its catalytic behavior [26]. The band at 988.7 cm^−1^ could be attributed to the V–O vibration of crystalline V_2_O_5_, and the bands at 800.5 cm^−1^ were associated with the W–O–W stretching of octahedrally coordinated W units [27,28,29]. The Raman spectra showed that the V_2_O_5_-WO_3_/TiO_2_ catalyst prepared by isotropic heating had high crystallinity, while that prepared by one-sided heating was mostly amorphous.

To further investigate the crystallinity of the prepared catalysts, we also compared the SAED patterns of the catalysts prepared using the one-sided heating (Figure 3a–c) and isotropic (Figure 3d–f) heating methods. In the SAED patterns, single spots only become visible when the beam is diffracted by a single crystal; however, amorphous materials yield ring patterns [30,31]. The diffraction patterns of V_2_O_5_ and WO_3_ prepared by the one-sided heating method were ring-shaped (Figure 3a–b), indicating amorphous structures [32]. However, those prepared following the isotropic heating method exhibited clear crystalline diffraction (Figure 3d,e). The TiO_2_ nanoparticles maintained their anatase structure, even after the application of heat treatment (Figure 3c,f). This indicates that the crystallinity of the catalyst was greatly affected by the heating conditions. That is, one-sided heating produced an amorphous V_2_O_5_-WO_3_/TiO_2_ catalyst, while isotropic heating produced a crystalline V_2_O_5_-WO_3_/TiO_2_ catalyst.

To identify the effect of the crystallinity of V_2_O_5_-WO_3_/TiO_2_ on its SCR performance, its NO_X_ removal efficiencies were measured in a fixed bed (Figure 4a–c). We found that the NO_X_ removal efficiency of the amorphous V_2_O_5_-WO_3_/TiO_2_ catalyst was negatively impacted at temperatures below 300 °C; however, it exceeded 94% at 300–400 °C. The crystalline V_2_O_5_-WO_3_/TiO_2_ catalyst achieved a NO_X_ removal efficiency of 82% at 200 °C; thus, it was 27% more efficient than the amorphous V_2_O_5_-WO_3_/TiO_2_ catalyst. Moreover, the efficiency increased to 100% in the temperature range of 240–400 °C (Figure 4a). NH_3_ conversion of amorphous and crystalline V_2_O_5_-WO_3_/TiO_2_ also showed a similar to the NO_X_ conversion value (Appendix A). Figure 4b shows the N_2_ selectivity of the V_2_O_5_-WO_3_/TiO_2_ catalysts at different temperatures. A trace amount of N_2_O in the amorphous and crystalline V_2_O_5_-WO_3_/TiO_2_ was generated at 350 °C, and the N_2_ selectivity of the amorphous and crystalline V_2_O_5_-WO_3_/TiO_2_ catalysts reached 73% and 81% from 500 °C, respectively. Figure 4c shows that SO_2_ affected the NO_X_ removal efficiency of the V_2_O_5_-WO_3_/TiO_2_ catalysts at 250 °C, which usually shows high deactivation caused by SO_2_. When SO_2_ gas was not added to the reactor, the NO_X_ removal efficiencies of the amorphous and crystalline V_2_O_5_-WO_3_/TiO_2_ were maintained at 80% and 99%, and then rapidly decreased to 69% and 89% with the introduction of SO_2_, respectively. However, it returned to 80% and 99% when the SO_2_ was removed. When SO_2_ was introduced to the reactor, SO_2_ gas directly reacts with V_2_O_5_-WO_3_/TiO_2_ catalysts, and it produces the ammonium sulfates [15]. Ammonium sulfates slowly block the active sites of V_2_O_5_-WO_3_/TiO_2_ catalysts, and it leads to the decrease of NOx removal efficiency. When SO_2_ is removed, the produced ammonium sulfates were gradually removed, and the V_2_O_5_-WO_3_/TiO_2_ catalysts can be regenerated and return to the initial condition. The crystalline V_2_O_5_-WO_3_/TiO_2_ showed slightly high resistance against SO_2_ compared with amorphous V_2_O_5_-WO_3_/TiO_2_. Additionally, Appendix A indicates that the crystalline V_2_O_5_-WO_3_/TiO_2_ catalyst showed higher NO_X_ removal efficiency than the amorphous V_2_O_5_-WO_3_/TiO_2_ catalyst in the temperature range of 150–500 °C under gas conditions containing SO_2_.

We conducted XPS, NH_3_-TPD, and H_2_-TPR analyses to further elucidate the effect of crystallinity on the NO_X_ removal performance of the V_2_O_5_-WO_3_/TiO_2_ catalysts, as shown in Figure 5. Figure 5a shows the survey peaks of the XPS results. The O 1s peaks can be fitted to two different peaks, i.e., chemisorbed oxygen (O_α_) at 530.9 eV and lattice oxygen (O_β_) at 530.1 eV [24]. Surface chemisorbed oxygen plays a critical role in the oxidation of NH_4_^+^ in SCR reactions as it is more mobile than lattice oxygen and promotes the oxidation of NO to NO_2_ [33,34]. Therefore, the presence of NO_2_ induces a “fast SCR” and the O_α_/(O_α_+O_β_) concentration ratio is the important value for the SCR reaction [33]. Figure 5b clearly indicates that the O_α_ ratio of the crystalline V_2_O_5_-WO_3_/TiO_2_ exceeded that of the amorphous V_2_O_5_-WO_3_/TiO_2_. V 2p was mainly composed of V^5+^ and V^4+^, and the two fitted peaks at 517.1 and 516.1 eV could be attributed to V^5+^ 2p_3/2_ and V^4+^ 2p_3/2_, respectively [35]. According to previous studies, V^4+^ can promote the adsorption of oxygen and form reactive oxygen species on the surface of a catalyst, leading to fast redox cycles and improving the redox properties [36]. Figure 5c shows that crystalline V_2_O_5_-WO_3_/TiO_2_ contains a higher proportion of V^4+^ than amorphous V_2_O_5_-WO_3_/TiO_2_. The V^4+^/(V^4+^+V^5+^) ratios of the crystalline and amorphous V_2_O_5_-WO_3_/TiO_2_ were 0.38 and 0.23, respectively (Table 3). Additionally, the W 4f on the surface of the catalyst was mainly composed of W 4f_7_ and W 4f_5_, while the Ti 3p was centered at 35.76, 38.12, and 37.50 eV, with a hexavalent state in the form of WO_3_ [37,38]. The W 4f XPS results of V_2_O_5_-WO_3_/TiO_2_ did not differ significantly, as shown in Figure 5d.

Figure 5e shows the NH_3_-TPD results for the amorphous and crystalline V_2_O_5_-WO_3_/TiO_2_ catalysts, such as the effects of their structures on the contents and strengths of the surface acidic sites of the catalysts [39,40]. The NH_3_-TPD profile of the amorphous and crystalline V_2_O_5_-WO_3_/TiO_2_ significantly varied in the temperature range of 100–900 °C, in which NH_3_ desorption of 13.74 cm^3^/g and 16.97 cm^3^/g was measured, respectively (Table 3). The thermal conductivity detector (TCD) signals at the lower and higher temperatures were considered to be Brønsted and Lewis acid sites [35,41], and the concentration of desorbed NH_3_ of the crystalline V_2_O_5_-WO_3_/TiO_2_ catalyst was higher, indicating a higher capability for adsorption. According to these results, the crystalline active materials contained more Brønsted and Lewis acid sites. We also produced H_2_-TPR profiles to investigate the redox properties of the amorphous and crystalline V_2_O_5_-WO_3_/TiO_2_. (Figure 5f) The amorphous V_2_O_5_-WO_3_/TiO_2_ exhibited have three apparent reduction peaks centered at 387.9 °C, 466.5 °C, and 796.4 °C, which could be assigned to the co-reduction of V^5+^ to V^3+^, which corresponds to the surface vanadium species, reduction of W^6+^ to W^4+^, and reduction of W^4+^ to W^0^ in tungsten oxide [30,42]. However, the main reduction peaks of crystalline V_2_O_5_-WO_3_/TiO_2_ shifted to lower temperatures at 394.9 °C, 461.2 °C, and 780.7 °C, respectively, which could be because the higher crystallinity of the active materials reduced the large amount of NO_X_, which promoted the release of lattice oxygen to further reduce the vanadium and tungsten species [42]. Consequently, we can confirm that the crystalline V_2_O_5_-WO_3_/TiO_2_ catalysts exhibited enhanced performance when NH_3_ gas adsorption and the reduction of NO and NO_2_ gas increased.

## 4. Conclusions

In this study, amorphous and crystalline V_2_O_5_-WO_3_/TiO_2_ catalysts were synthesized following two different heating methods to investigate the effects of crystallinity on the acidity, surface species, and performance of the catalysts. The isotropic heating method formed crystalline V_2_O_5_ and WO_3_ structures that contained more Brønsted and Lewis acid sites. The crystalline V_2_O_5_-WO_3_/TiO_2_ catalyst also had higher chemisorbed oxygen and V^4+^ species ratios than the amorphous catalyst. The crystalline structure of the V and W species significantly enhanced the SCR reactions on the surface of the catalysts, resulting in high NO_X_ removal efficiency and N_2_ selectivity over a wide temperature range of 200–450 °C. These results may contribute to future studies on SCR catalysts and other catalyst systems.

## Figures and Tables

**Figure 1 nanomaterials-11-01452-f001:**
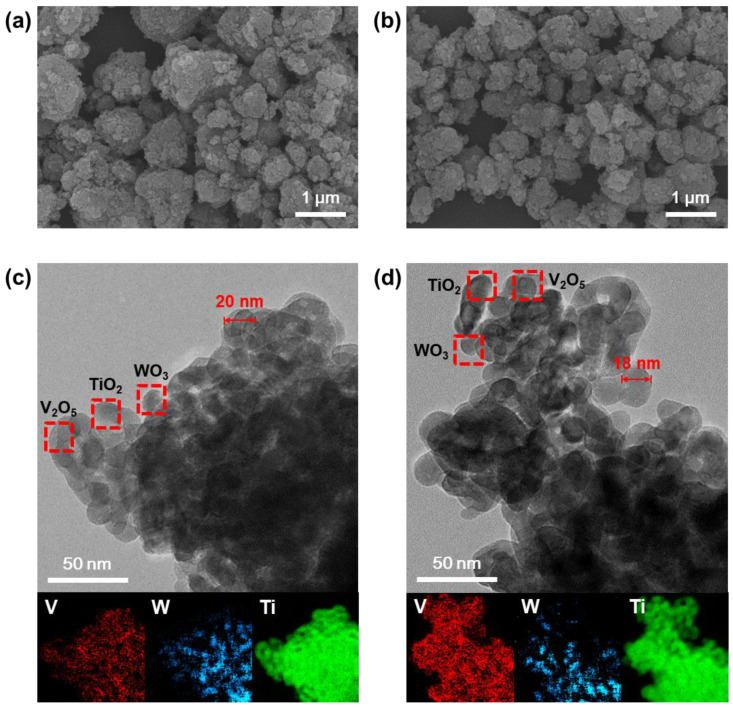
SEM images of the V_2_O_5_-WO_3_/TiO_2_ prepared by the (**a**) one-sided heating and (**b**) isotropic heating methods. TEM images of the V_2_O_5_-WO_3_/TiO_2_ prepared by the (**c**) one-sided heating and (**d**) isotropic heating methods (insets show the EELS elemental mapping of V, W, and Ti).

**Figure 2 nanomaterials-11-01452-f002:**
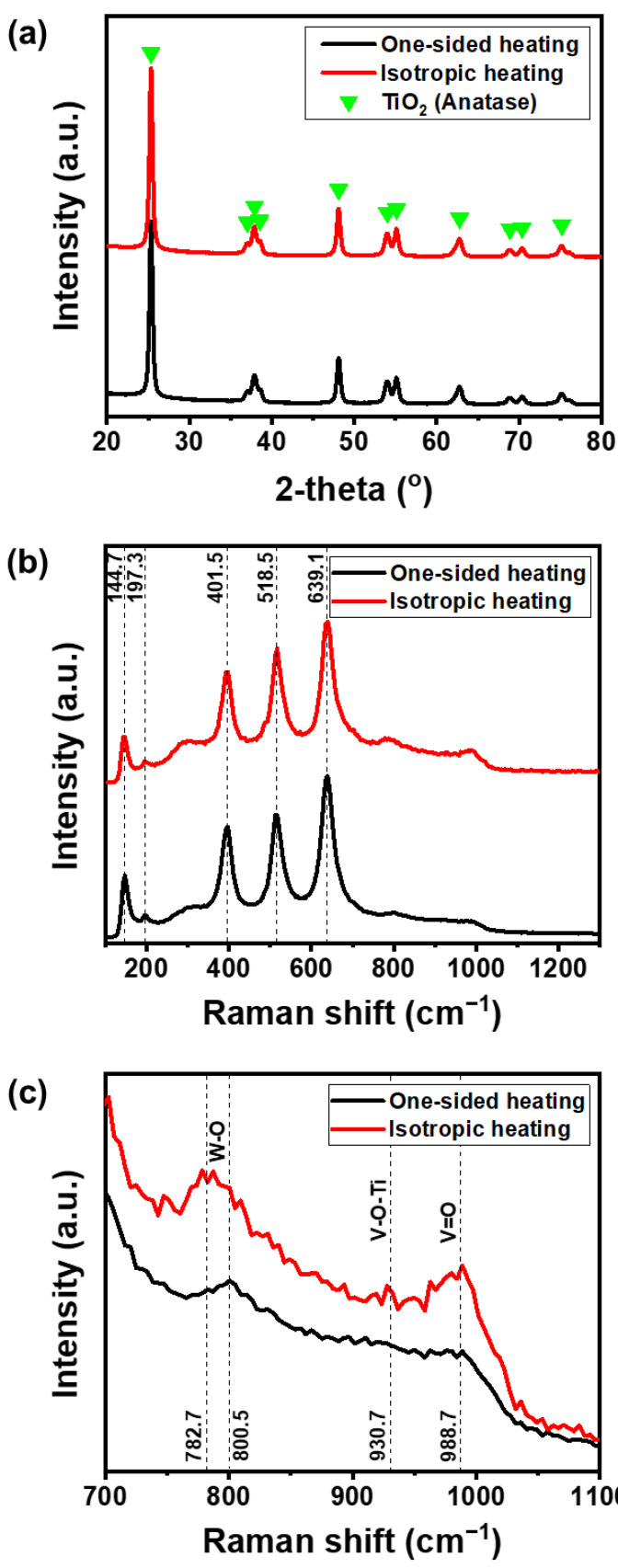
(**a**) XRD patterns and (**b**,**c**) Raman spectra of V_2_O_5_-WO_3_/TiO_2_ with different structures (black and red lines represent the V_2_O_5_-WO_3_/TiO_2_ prepared by the one-sided heating and isotropic heating methods, respectively).

**Figure 3 nanomaterials-11-01452-f003:**
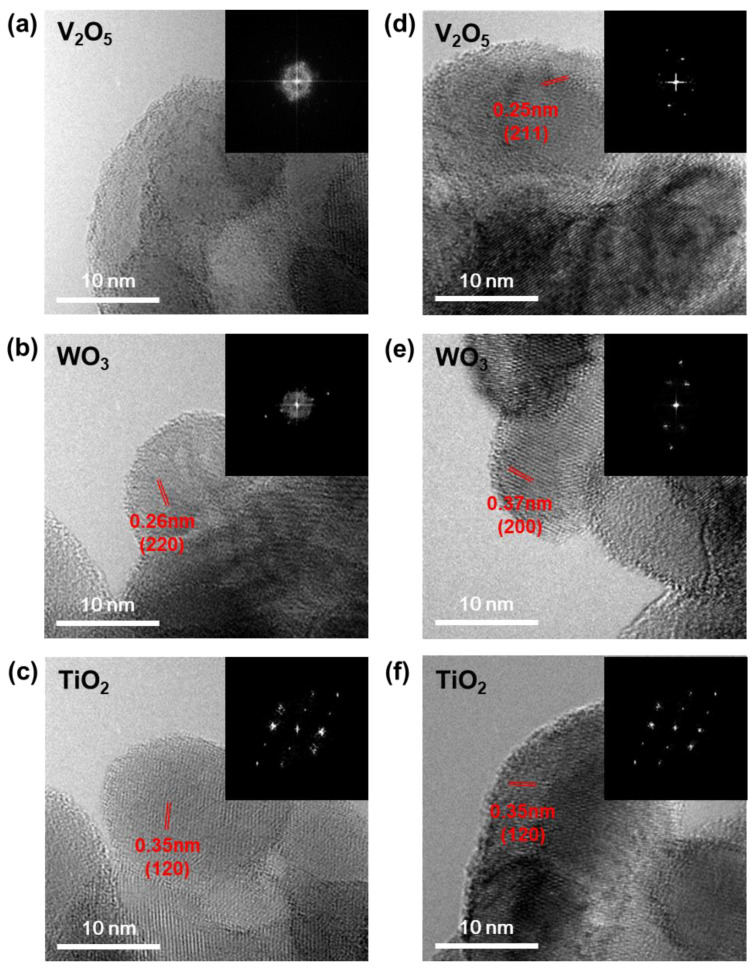
TEM images and SAED patterns (inset) of the V_2_O_5_-WO_3_/TiO_2_ prepared by (**a**–**c**) one-sided heating and (**d**–**f**) isotropic heating methods.

**Figure 4 nanomaterials-11-01452-f004:**
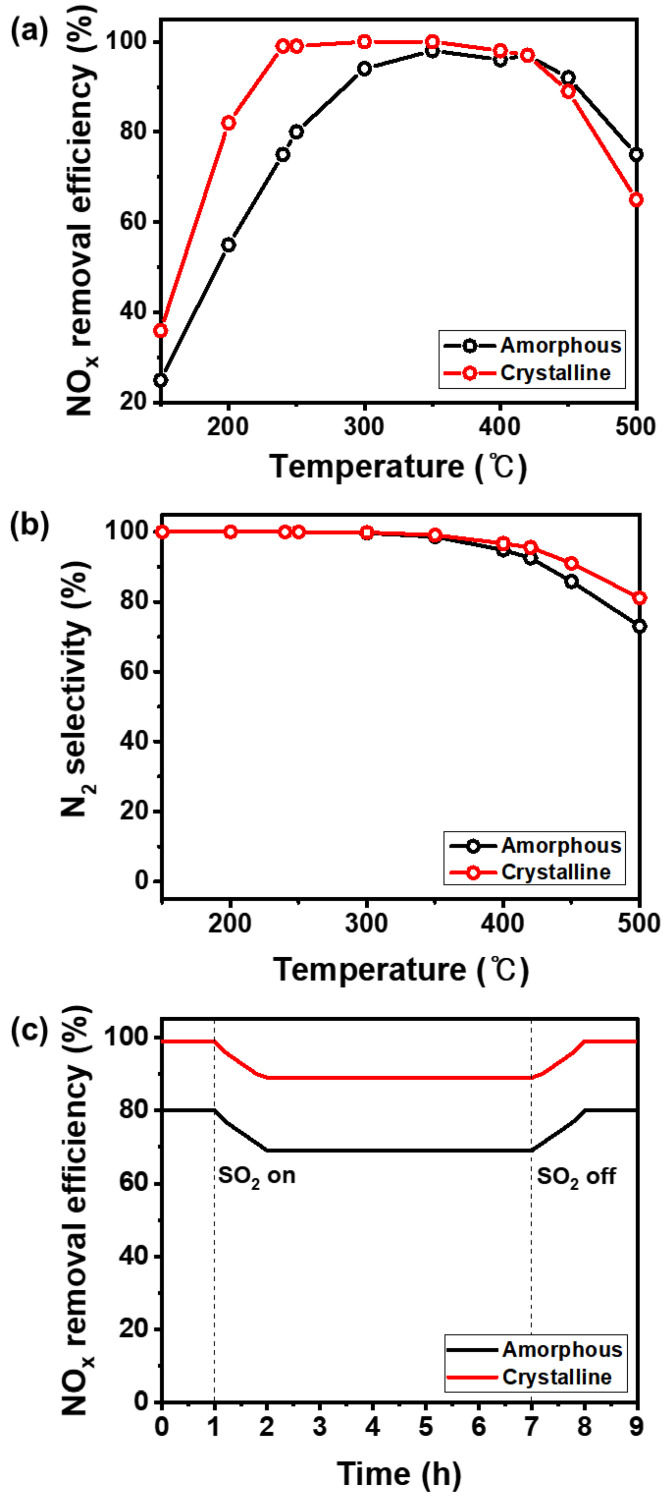
(**a**) NOx removal efficiency; (**b**) N_2_ selectivity of the V_2_O_5_-WO_3_/TiO_2_ catalysts and (**c**) SO_2_ tolerance of the V_2_O_5_-WO_3_/TiO_2_ catalysts with different crystal structures at 250 °C (black and red lines represent the amorphous and crystalline V_2_O_5_-WO_3_/TiO_2_, respectively). Reaction conditions: [NO] & [NH_3_] = 300 ppm, [O_2_] = 5 vol.%, [GHSV] = 60,000 h^−1^.

**Figure 5 nanomaterials-11-01452-f005:**
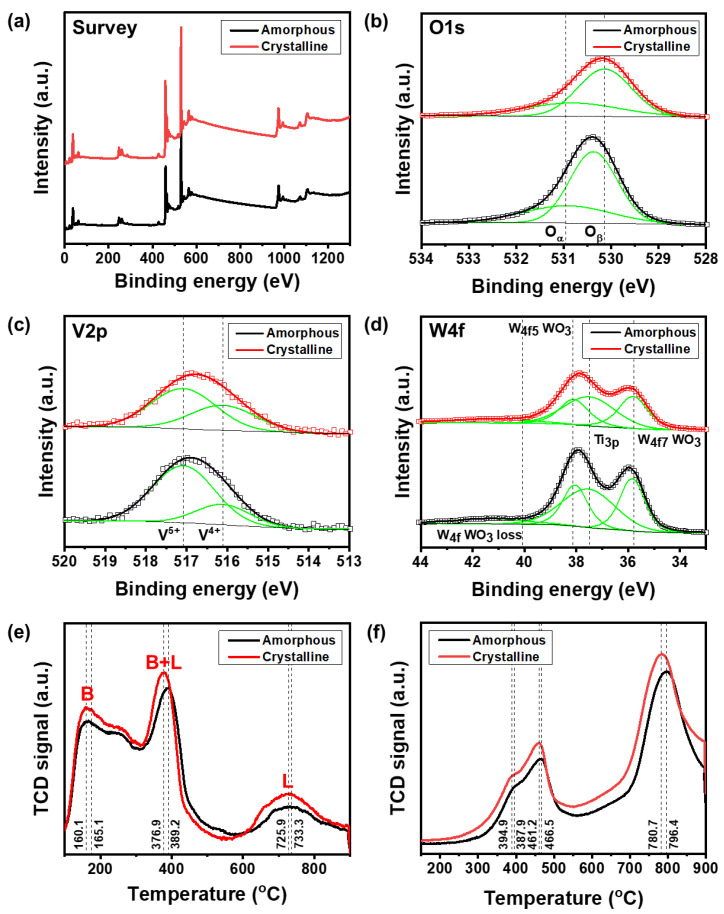
XPS spectra for (**a**) survey, (**b**) O1s, (**c**) V2p, and (**d**) W4f of amorphous V_2_O_5_-WO_3_/TiO_2_ and crystalline V_2_O_5_-WO_3_/TiO_2_ (**e**) NH_3_-TPD profiles of amorphous V_2_O_5_-WO_3_/TiO_2_ and crystalline V_2_O_5_-WO_3_/TiO_2_. B and L indicate Brønsted and Lewis acid sites, respectively. (**f**) H_2_-TPR profiles of amorphous V_2_O_5_-WO_3_/TiO_2_ and crystalline V_2_O_5_-WO_3_/TiO_2_.

**Table 1 nanomaterials-11-01452-t001:** Brunauer–Emmet–Teller (BET) results of the V_2_O_5_-WO_3_/TiO_2_ prepared by the one-sided heating and isotropic heating methods.

Sample	BET Surface Area; S_BET_ (m^2^/g)	Pore Volume (cm^3^/g)	Pore Size (nm)
One-sided	69.6	0.252	14.48
Isotropic	70.2	0.257	14.67

**Table 2 nanomaterials-11-01452-t002:** X-ray fluorescence analysis of the V_2_O_5_-WO_3_/TiO_2_ prepared by the one-sided heating and isotropic heating methods.

Sample	TiO_2_	WO_3_	V_2_O_5_	SO_3_	SiO_2_
One-sided	86.92	10.19	2.02	0.70	0.17
Isotropic	87.05	10.04	2.03	0.66	0.22

**Table 3 nanomaterials-11-01452-t003:** The ratio of Oα, V^4+^ of amorphous and crystalline V_2_O_5_-WO_3_/TiO_2_ measured by XPS, NH_3_-temperature-programmed desorption and H_2_-temperature-programmed reduction integral intensity of amorphous and crystalline V_2_O_5_-WO_3_/TiO_2_.

Sample	O_α_/(O_α_ + O_β_)	V^4+^/(V^4+^ + V^5+^)	NH_3_ Desorption (cm^3^/g)	H_2_ Consumption (cm^3^/g)
Amorphous	0.30	0.23	13.74	40.17
Crystalline	0.33	0.38	16.97	46.06

## Data Availability

Data are contained within the article.

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
