# Peer review of "Effect of Catalyst Crystallinity on V-Based Selective Catalytic Reduction with Ammonia"

_nanomaterials, 2021, doi:10.3390/nano11061452_

Round 1

Reviewer 1 Report

This paper reported V2O5-WO3/TiO2 catalysts with different crystallinities via one-sided and isotropic heating methods for selective catalytic reduction with ammonia. The isotropic heating method produced crystalline V2O5 and WO3, promoting the SCR reactions and resulting in high NOx conversion and N2 selectivity. The study was interesting and systematic. However, several issues have to be addressed before I can recommend publication.

  • Line 46-47. It is well known that, the thermal stability is a big problem for V2O5-based catalysts. Thus, it is not correct for the statement of “its excellent thermal stability at temperatures exceeding 300 °C”.
  • The equation for the calculation of N2 selectivity is incorrect.
  • In NH3-TPD profiles, is there the desorption of physiosorbed NH3 species at low temperature? DRIFTs of NH3 adsorption can be used to differentiate the B and L acid sites.
  • In Page 3, ‘the V2O5 and WO3 were uniformly distributed on the TiO2 supports with no agglomeration’. It conflicts with Figure 1 and 3.
  • Some more recent papers on V2O5-based NH3-SCR would be helpful for giving the readers a full picture of state-of-the-art catalysts, such as Chemical Communications, 2021, 57: 355-358; Catalysts, 2020, 10, 1421, doi: 10.3390/catal10121421.

Author Response

We thank the reviewer for his/her comments on our manuscript. Based on the reviewer’s comments, we have now made a few changes to the manuscript, which are uploaded as a separate file.

Reviewer 2 Report

This paper meets the criteria of novelty and wide interest for the publication on Nanomaterials we suggested some revision to further improve the work impact.

In this manuscript, the Authors applied two different synthetic procedures for the preparation of V2O5-WO3/TiO2 systems for catalytic purposes. They show how the isoform drying lead to a more crystalline material with superior performances. We believe the work is interesting with soundly characterisations and discussions, so it deserves publication on Nanomaterials. We have only a question for the authors: they tested the effect of SO2 addition to the reactor, why they believe the crystalline materials should be less affected as it seems from the results to this “pollutant” ? this finding maybe deserve further discussion.

Author Response

(The authors gave the same response as above.)
